# Interaction of High Temperature Stress and *Wolbachia* Infection on the Biological Characteristic of *Drosophila melanogaster*

**DOI:** 10.3390/insects14060558

**Published:** 2023-06-15

**Authors:** Die Hu, Wanning Li, Ju Wang, Yaqi Peng, Yueli Yun, Yu Peng

**Affiliations:** 1Hubei Key Laboratory of Regional Development and Environmental Response, Faculty of Resources and Environmental Science, Hubei University, Wuhan 430062, China; 2State Key Laboratory of Biocatalysis and Enzyme Engineering, School of Life Sciences, Hubei University, Wuhan 430062, China

**Keywords:** body weight, body length, developmental durations, survival rate, interaction effect

## Abstract

**Simple Summary:**

This study explored the interaction effects of high temperature stress and *Wolbachia* infection on the biological characteristic of fruit fly *Drosophila melanogaster*. Results showed that high temperature and *Wolbachia* infection had a two-way interaction effect on hatching rate, developmental durations, emergence rate, body weight and body length of F_1_, F_2_ and F_3_ flies, and the interaction effect also existed on oviposition amount of F_3_ flies, and on pupation rate of F_2_ and F_3_ flies. The study also showed that high temperature stress deceased the *Wolbachia* vertical transmission efficiency between generations, and both high temperature stress and *Wolbachia* infection can reduce the survival of *D. melanogaster*. The results indicated that high temperature stress and *Wolbachia* infection had negative effects on the growth and development of *D. melanogaster*.

**Abstract:**

It was reported that temperature affects the distribution of *Wolbachia* in the host, but only a few papers reported the effect of the interaction between high temperature and *Wolbachia* on the biological characteristic of the host. Here, we set four treatment *Drosophila melanogaster* groups: *Wolbachia*-infected flies in 25 °C (W^+^M), *Wolbachia*-infected flies in 31 °C (W^+^H), *Wolbachia*-uninfected flies in 25 °C (W^-^M), *Wolbachia*-uninfected flies in 31 °C (W^-^H), and detected the interaction effect of temperature and *Wolbachia* infection on the biological characteristic of *D. melanogaster* in F_1_, F_2_ and F_3_ generations. We found that both temperature and *Wolbachia* infection had significant effects on the development and survival rate of *D. melanogaster*. High temperature and *Wolbachia* infection had interaction effect on hatching rate, developmental durations, emergence rate, body weight and body length of F_1_, F_2_ and F_3_ flies, and the interaction effect also existed on oviposition amount of F_3_ flies, and on pupation rate of F_2_ and F_3_ flies. High temperature stress reduced the *Wolbachia* vertical transmission efficiency between generations. These results indicated that high temperature stress and *Wolbachia* infection had negative effects on the morphological development of *D. melanogaster*.

## 1. Introduction

*Wolbachia* is a kind of Gram-negative endosymbiotic bacteria that is widely distributed in arthropods and nematodes [1]. Weinert et al. reported in 2015 that approximately 52% of arthropods were infected with *Wolbachia* [2]. *Wolbachia* spreads within and between host species through vertical transmission and horizontal transmission. Vertical transmission refers to the transmission from oocyte to offspring through reproduction [3,4]. The horizontal transmission of *Wolbachia* can occur between different individuals of the same host, as well as between different species of host [5]. Reproductive manipulation of *Wolbachia* on its hosts is one of the reasons why it was widely studied in recent years [6,7]. *Wolbachia* infection is associated with a variety of reproductive anomalies in the host, including cytoplasmic incompatibility [8,9], parthenogenesis induction [10], male sterility [7,11], feminization [12,13] and fertility modification [14,15]. In addition to reproductive regulation effects, *Wolbachia* also has many other impacts on the host. For example, the *Wolbachia* strain *wMel* can change the composition of gut commensal bacteria in *Drosophila melanogaster* [16]. The fruit flies infected with *Wolbachia* are less susceptible to mortality induced by a range of RNA viruses [17]. *Wolbachia* infection significantly decreases developmental duration in a bean beetle *Callosobruchus chinensis* [18], but significantly increases development time in the cabbage root fly *Delia radicum* [19]. Su et al. reported that *Wolbachia* infection in a small spider *Hylyphantes graminicola* (Linyphiidae) can shorten the developmental duration of the host [20]. Hickin et al. reported that *Cimex lectularius* (common bed bug) had nutritional dependence on their *Wolbachia* (*Wb*) symbiont, and the authors verified that *Wb* can provide B-vitamins for this bed bug through adding B-vitamin to the blood meal of aposymbiotic bedbugs [21].

Temperature is a key environmental modulator of host–pathogen interactions, which constrains the rate of biological reactions and sets limits to performance and survival [22,23]. Temperature also plays a fundamental role in the dynamics of host–pathogen interactions [24]. The thermal tolerance of an organism limits its ecological and geographic ranges and is potentially affected by dependence on temperature-sensitive symbiotic partners [25]. High temperature may also be one of the reasons for the incomplete symbiosis of individuals in the same population in nature [3,26]. Corbin et al. outlined the evidence that temperature impacts of ecologically contingent benefits on natural infections of insects and reproductive manipulation phenotypes of symbionts on hosts are altered by thermal environment. They also provided evidence with respect to temperature impacts upon vertical transmission and the direct physiological cost of symbiont infection [27]. Endosymbionts, which rely on their hosts for nutrition, can impose a cost on their host when the host is under physiological stress. For example, uninfected heat-shocked aphids were 24% more likely to survive to adulthood than infected heat-shocked aphids, and infected heat-shocked aphids also suffered higher sterility rates [28]. A study of *Wolbachia*-infected *D. melanogaster* also indicated thermal impacts on the cost of carrying a symbiont [29].

So far, extensive research was conducted on the interaction between symbiotic bacteria and their hosts, as well as the interaction between temperature and symbiotic bacteria. Only a few papers studied the interaction between *Wolbachia* and temperature on hosts. For example, Strunov et al. analyzed the effects of *Wolbachia* infection, *Wolbachia* variants, the host’s sex and temperature on developmental life-history traits, lifespan and fecundity of *D. melanogaster*, and the results indicated that temperature and infection had highly significant effects on fecundity, and significant interactions were found between temperature and infection. They also found significant two-way interactions between temperature and infection on longevity of *D. melanogaster* and no interactions existed between infection and temperature on juvenile development time [30]. However, earlier reports showed that temperature had a major impact on development time and body size of *D. melanogaster* [31]. By contrast, a previous study [32] also indicated that there were no direct effects of *Wolbachia* infection nor variant, nor interactions with temperature on juvenile development time. To further explore the interaction between temperature and *Wolbachia* on the development of *D. melanogaster*, in this study, we set four treatment group flies: *Wolbachia*-infected flies cultured in 25 °C (W^+^M), *Wolbachia*-infected flies in 31 °C (W^+^H), *Wolbachia*-uninfected flies in 25 °C (W^-^M), *Wolbachia*-uninfected flies in 31 °C (W^-^H), and detected oviposition amount, hatching rate, developmental duration, pupation rate, emergence rate, adult body weight, adult body length, adult lifespan and survival rate in each group, and we then analyzed the interaction between high temperature stress and *Wolbachia infection* on *D. melanogaster*.

## 2. Materials and Methods

### 2.1. Fruit Fly Collection and Isofemale Lines Establishing

*D*. *melanogaster* were collected in August 2018 from Shahu park (Wuhan, China), which were identified based on morphology. Gravid female flies were reared individually on standard treacle–semolina–yeast–agar media in an artificial climate incubator (25 ± 0.5 °C, RH: 40 ± 5%, L: D = 14:10). After each female gravid fly laid eggs, the female adult flies were then used to extract DNA for *Wolbachia* infection screening and *Wolbachia* strain diagnosis. DNA of single flies were extracted using DNA extraction kit (CWBio, China). *Wolbahica* infection status of *D*. *melanogaster* was detected with *Wolbachia*-specific primers and conditions as described by Braig et al. (1998) [33]. *Wolbachia* strain of *D*. *melanogaster* were diagnosed through amplifying a variable tandem repeat region of the *Wolbachia* genome (VNTR-141) [30,34]. Then, we confirmed the fly strain was *wMel* variants, and isofemale lines of *wMel* D. melanogaster (W+) were established in laboratory. Isofemale lines of *Wolbachia*-uninfected flies (W^-^) was subsequently generated by tetracycline treatment following established protocol [35] and confirmed to be *Wolbachia*-free by PCR using *Wolbachia* surface protein gene (*wsp*).

### 2.2. Growth, Development and Lifespan of W^+^ and W^-^ Flies in High Temperature Stress

We selected W^+^ and W^-^ flies from the isofemale lines reared in laboratory, and 4 treatment combinations were set: W^+^ flies cultured in 25 °C (W^+^M), W^+^ flies in 31 °C (W^+^H), W^-^ flies in 25 °C (W^-^M), W^-^ flies in 31 °C (W^-^H). In this study, through preliminary experimental testing, the optimal high temperature for the survival of *D. melanogaster* was 31 °C. Therefore, 31 °C was used as the temperature value for high temperature stress in this experiment.

Virginal fruit flies (including female and male individuals) were collected individually within 8 h after emergence from F_1_ generation of W^+^M, W^+^H, W ^m^ and W^-^H populations, respectively, and reared with standard treacle–semolina–yeast–agar media. When the virginal flies were at the 5-day-old stage, five flies each (female:male = 3:2) were selected randomly and were put into a Petri dish with standard treacle–semolina–yeast–agar media and cultured in artificial climate incubator (W^+^M and W ^m^ for 25 ± 1 °C, W^+^H and W^-^H for 31 ± 1 °C, RH 40 ± 5%, L: D = 14:10); After 6 h, the female and male adult flies were removed, and the number of eggs were recorded with a magnifying glass. Then, we calculated the hatching rate of each treatment, respectively, after 24 h. Then, the number of pupae, the number of emergence and the time from egg hatching to emergence were recorded successively. There were 12 biological replicates for each treatment.

A total of 20 unmated female flies and 20 unmated male flies were selected within 8 h after emergence from F_1_ generation of W^+^M population. The same sampling method was applied to the F_1_ generation of W^-^M, W^+^H and W^-^H populations. Then, the body length and the weight of the flies from four treatments were measured (Leica Microscopes and Electronic precision balance).

A total of 30 unmated flies were selected within 8 h after emergence from F_1_ generation of W^+^M population and reared individually on standard treacle–semolina–yeast–agar media in an artificial climate incubator (25 ± 0.5 °C, L:D = 14:10, RH: 40 ± 5%). The same sampling method was applied to the F_1_ generation of W ^m^ (cultured in 25 ± 0.5 °C), W^+^H (cultured in 31 ± 0.5 °C) and W^-^H (cultured in 31 ± 0.5 °C) populations. The number of dead flies from four treatments was recorded, respectively, every 24 h.

Gravid female flies from F_1_ generation of W^+^M, W^+^H, W ^m^ and W^-^H populations were reared individually on standard treacle–semolina–yeast–agar media in an artificial climate incubator (25 ± 0.5 °C, RH: 40 ± 5%, L: D = 14:10). After these gravid female individuals laid eggs, which then developed into adult flies through larvae and pupae stages, we obtained the F_2_ generation fruit flies. We obtained F_3_ generation flies of W^+^M, W^+^H, W ^m^ and W^-^H populations from gravid female flies of F_2_ generation. The growth and development characteristics and lifespan of F_2_ and F_3_ generation flies of W^+^M, W^+^H, W ^m^ and W^-^H populations were also recorded using the same method as those of the F_1_ generation.

### 2.3. Wolbachia Vertical Transmission Efficiency in D. melanogaster under Continuous High Temperature Stress

RNA for each treatment was extracted by Trizol Method, and cDNA was obtained by reverse transcription kit (Yeasen, Shanghai, China). qPCR was conducted by SYBR Green qPCR mix (Monad, Wuhan, China) in real-time fluorescence quantitative PCR instrument (Agilent, Santa Clara, CA, USA). The primers were designed by DNAStar software according to the sequences of actin reference genes of *D*. *melanogaster* published in the Genebank database. We used the expression of *wsp* (*wsp*-F: 5′-CTG GTG GTG GTG CAT TTG GT-3′; *wsp*-R: 5′-CCA ACG TAT GGA GTG ATA GGC A-3′) to detect the density of *Wolbachia*. The *β*-actin gene was used as a reference sequence (actin-F: 5′-AAC ACC ATC GAA CCA CTC CC-3′; actin-R: 5′-ACA TCA GCG AGC TTG GCT TT-3′). RT-qPCR was performed according to the method of Livak and Schmittgen [36]. The 2^−ΔΔCt^ method was used to calculate the relative expression of genes [36]. We detected the *Wolbachia* vertical transmission efficiency through relative expression of *wsp* (*Wolbachia* surface protein gene) in F_1_, F_2_ and F_3_ generation of W^+^M and W^+^H flies. Five unmated W^+^M flies (within 6 h after emergence) were pooled for a sample for RNA extraction in F_1_, F_2_ and F_3_ generation, and five unmated W^+^H flies (within 6 h after emergence) were pooled for RNA extraction in each generation. There were three biological replicates per generation.

### 2.4. Data Processing

We first used two-way ANOVA to analyze the interaction between temperature and *Wolbachia* infection. If there was an interaction between the two factors, Tukey HSD was used for multiple comparisons. If there was no interaction between the two factors, Student’s *t*-test was used for pairwise comparisons. Three-way ANOVA was used to analyze the interaction among temperature, *Wolbachia* infection and host’s sex. The survival curves were analyzed using the Kaplan–Meier method (log-rank test). SPSS (26.0) and Graph Pad Prism7 were used for data analysis.

## 3. Results

### 3.1. Effect of High Temperature and Wolbachia Infection on the Growth and Development of D. melanogaster

In F_1_ generation *D. melanogaster* flies, we found that temperature and *Wolbachia* infection had no interaction effect on oviposition amount and pupation rate (two-way ANOVA, F_1,44_ = 0.493, *p* = 0.486; F_1,44_ = 1.599, *p* = 0.213), but there was interaction effect on hatching rate, developmental durations and emergence rate (two-way ANOVA, F_1,44_ = 4.980, *p* = 0.031; F_1,84_ = 51.016, *p* < 0.0001; F_1,44_ = 216.342, *p* < 0.0001) (see Figure 1). W ^m^ fly group showed the highest hatching rate and emergence rate. Compared with W ^m^ group, the hatching rates and emergence rates were significantly lower in W^+^M and W^-^H groups (*p* < 0.05). Compared with W^+^M and W^-^H groups, the hatching rates and emergence rates were significantly lower in W^+^H group (*p* < 0.05) (Figure 1B,E), which indicated that *Wolbachia* infection and high temperature had a positive interaction effect. In addition, W^+^M fly group had the longest developmental duration, while W^+^H fly group had the shortest developmental duration (Figure 1C), which indicated that high temperature reduced the developmental duration in the case of *Wolbachia* infection. The oviposition amount of W^+^ flies was significantly lower compared to the W^-^ flies at the same temperature (Student’s *t*-test, *p* < 0.001), and the oviposition amount of flies in 31 °C was significantly higher than that of the flies in 25 °C under the same *Wolbachia* infection status (Student’s *t*-test, *p* < 0.001) (Figure 1A). The pupation rate of flies at 31 °C was lower than that of flies at 25 °C in both *Wolbachia*-infected and uninfected states, and the pupation rate of *Wolbachia*-infected flies was lower than that of *Wolbachia*-uninfected flies under both 25 °C and 31 °C rearing conditions (Student’s *t*-test, *p* < 0.001) (Figure 1D). An analysis of the *t*-test showed that *Wolbachia* infection and high temperature had an effect on the oviposition amount and pupation rates of F_1_ generation *D. melanogaster*.

In F_2_ generation flies (see Figure 2), we found that temperature and *Wolbachia* infection had an interaction effect on hatching rate, developmental durations, pupation rate and emergence rate (two-way ANOVA, F_1,44_ = 19.314, *p* = 0.0001; F_1,84_ = 4.003, *p* = 0.049; F_1,44_ = 25.252, *p* < 0.0001; F_1,44_ = 172.458, *p* < 0.0001). W^-^M fly group showed the highest hatching rate, pupation rate and emergence rate. Compared with W^-^M group, the hatching rates, pupation rate and emergence rates were significantly lower in W^+^M, W^-^H and W^+^H groups (*p* < 0.05). In addition, compared with W^-^M group, the developmental duration was longer in W^+^H, W^-^H and W^+^M groups (*p* < 0.05). The oviposition amount of W^+^ flies was significantly lower compared to the W^-^ flies at the same temperature (Student’s *t*-test, *p* < 0.001). The oviposition amount of flies in 31 °C was significantly lower than that of the flies in 25 °C under the same *Wolbachia* infection status (Student’s *t*-test, *p* < 0.001) (Figure 2A).

In F_3_ generation flies (see Figure 3), we found that temperature and *Wolbachia* infection had an interaction effect on oviposition amount, hatching rate, developmental durations, pupation rate and emergence rate (two-way ANOVA, F_1,44_ = 4.743, *p* = 0.035; F_1,44_ = 8.958, *p* = 0.005; F_1,84_ = 5.415, *p* = 0.022; F_1,44_ = 132.645, *p* < 0.0001; F_1,44_ = 41.705, *p* < 0.0001). W^-^M fly group showed the highest oviposition amount, hatching rate, pupation rate and emergence rate. Compared with W^-^M group, the oviposition amount, hatching rates, pupation rate and emergence rates were significantly lower in W^+^M, W^-^H and W^+^H groups (*p* < 0.05). In addition, compared with W^-^M group, the developmental duration was longer in W^+^H, W^-^H and W^+^M groups (*p* < 0.05).

We found a significant interaction between *Wolbachia* infection and temperature on the effect of the body weight in F_1_, F_2_ and F_3_ generation flies (F_1,152_ = 9.161, *p* = 0.003; F_1,152_ = 9.559, *p* = 0.002; F_1,152_ = 35.612, *p* < 0.001). The interaction between *Wolbachia* infection and sex on the body weight of F_1_ generation flies was found (F_1,152_ = 23.904, *p* < 0.001), and temperature and sex had an interaction effect on the body weight of F_2_ and F_3_ generation flies (F_1,152_ = 361.633, *p* < 0.001; F_1,152_ = 410.561, *p* < 0.001). There was a significant interaction effect among *Wolbachia* infection, temperature and sex on the weight of F_3_ generation flies (three-way ANOVA, F_1,152_ = 6.652, *p* = 0.011) (See Table 1). W^-^M fly group had the longest body weight in F_1_, F_2_ and F_3_ generation (including female and male flies), while W^+^H fly group had the lowest body weight in F_1_, F_2_ and F_3_ generation (including female and male flies) (Figure 4).

We found a significant interaction effect between *Wolbachia* infection, temperature and sex on the body length of F_1_, F_2_ and F_3_ generation flies (three-way ANOVA, F_1,152_ = 26.590, *p* < 0.011; F_1,152_ = 28.304, *p* < 0.001; F_1,152_ = 40.078, *p* < 0.001) (See Table 2). W^-^M fly group had the longest body length in F_1_, F_2_ and F_3_ generation (including female and male flies), while W^+^H fly group had the lowest body length in F_1_, F_2_ and F_3_ generation (including female and male flies) (Figure 5).

### 3.2. Effect of High Temperature Stress and Wolbachia Infection on the Survival Rate of D. melanogaster

Compared with *Wolbachia* negative flies, the survival rate of *Wolbachia* positive flies from F_1_, F_2_ and F_3_ generation at the same temperature were significantly decreased (log-rank test, * *p* < 0.05) (Figure 6). Compared with flies reared at 25 °C, the survival rate of flies reared at 31 °C from F_1_, F_2_ and F_3_ generation under the same *Wolbachia* infection status were significantly decreased (log-rank test, * *p* < 0.001) (Figure 6). The results indicate that both high temperature stress and *Wolbachia* infection can reduce the survival of *D. melanogaster.*

### 3.3. Effect of High Temperature Stress on Wolbachia Vertical Transmission Efficiency

The relative expression of *wsp* of the W^+^ flies reared at 31 °C from F_1_ generation to F_3_ generation decreased gradually (one-way ANOVA, *p* < 0.05), while there was no significant difference of *wsp* expression among F_1_, F_2_ and F_3_ generations of W^+^ flies reared at 25 °C (one-way ANOVA, *p* > 0.05) (Figure 7). The result showed that high temperature stress deceased the *Wolbachia* vertical transmission efficiency from F_1_ to F_3_ generation of *D. melanogaster*.

## 4. Discussion

*Wolbachia* are widely distributed in arthropods, and approximately 52% of arthropods were infected with *Wolbachia* [2] In recent years, there were many studies on the effects of *Wolbachia* on host reproduction and fitness [37,38], and the effects of temperature on the distribution and abundance of *Wolbachia* in hosts were also reported widely [2,39]. So far, extensive research was conducted on the interaction between endosymbionts and their hosts, as well as the interaction between temperature and endosymbionts. Only a few papers studied the interaction between *Wolbachia* and temperature on hosts [30]. In this study, we set four treatment group flies (W^+^M, W^+^H, W^-^M and W^-^H) to detect the difference of the growth, development, adult lifespan and *Wolbachia* distribution of *D. melanogaster* in F_1_, F_2_ and F_3_ generations, and tested the interaction of temperature and *Wolbachia* infection on the biological characteristics of host. The result showed that temperature stress and *Wolbachia* infection had interaction on the growth and development of flies.

Our results showed that both high temperature stress (31 °C) and *Wolbachia* infection affected the fecundity and development of *D. melanogaster*. High temperature stress and *Wolbachia* infection showed an interaction on the fecundity and development of *D. melanogaster*. Specifically, the short-term high temperature stress (stress in F_1_ generation) increased the oviposition amount and shortened the pupation rate of *D. melanogaster*, which seemed to have a positive impact on fecundity of the host. However, long-term interaction between temperature and *Wolbachia* infection (in F_1_, F_2_ and F_3_ generation) can reduce the oviposition amount, hatching rate, pupation rate, emergence rate and prolong the development duration, which is a negative effect on the host. Strunov et al. [30] studied the impact of *Wolbachia* and temperature interactions on the host in the short term high temperature stress, and they found that the development time, femur lengths and the total number of ovarioles were significantly affected by higher temperature in F_1_ generation *D*. *melanogaster*; moreover, both *Wolbachia* and temperature had effects on longevity and female fecundity of F_1_ generation *D*. *melanogaster*, and significant two-way interactions between temperature and *Wolbachia* infection were found on fecundity and longevity of F_1_ generation. In our study, we not only detected the two-way interactions between temperature and *Wolbachia* infection on the F_1_ generation *D*. *melanogaster*, but we also tested the interaction between temperature and *Wolbachia* infection on the F_2_ and F_3_ generation. These results indicated that high temperature stress and *Wolbachia* infection had negative effects on the morphological development of *D. melanogaster*. This may be a morphological adaptation to extreme temperature. Research showed significant differences between the survival curves of infected flies and uninfected controls at both 25 °C and 29 °C [39]. In this study, we tested the survival curves of infected and uninfected flies at 25 °C and 31 °C in F_1_, F_2_ and F_3_ generation, and we found that both high temperature stress and *Wolbachia* infection can reduce the survival of *D. melanogaster.*

Research showed that *Wolbachia*-infected hosts prefer cooler temperatures and might be likely to seek out cooler microclimates, which would reduce exposure to higher temperature and lessen the fitness consequences of high temperatures [40]. *D*. *melanogaster* exhibits strong circadian and neurally controlled temperature preference behavior, which centers around 24–27 °C [41,42]. The research of Arnold et al. (2019) [40] showed that *Wolbachia*-infected flies prefer a cooler mean temperature than uninfected flies. Therefore, we suspect that *Wolbachia*-infected flies may be able to reduce the survival pressure caused by heat stress, reduce the impact of high temperature on their growth and development and reduce their own heat generation.

Endosymbionts rely on their hosts for nutrition, and they can impose a cost on their host [43,44]. Costs may manifest when the host is under physiological stress [27]. There were a few studies examining the physiological cost of symbionts at different temperatures. Russell and Moran (2006) [28] reported that uninfected heat-shocked aphids were 24% more likely to survive to adulthood than infected heat-shocked aphids. In this study, temperature and *Wolbachia* infection had an interaction effect on oviposition amount, hatching rate, developmental durations, pupation rate and emergence rate of F_3_ generation flies. Compared with W^-^M group, oviposition amount, hatching rate, pupation rate and emergence rate of F_3_ generation flies in W^+^H group significantly decreased, while the developmental durations in W^+^H group were significantly longer than that in W^-^M group. These results indicated that *Wolbachia* infection imposed a cost to fly hosts at 31 °C, which resulted in flies laying fewer eggs compared to the W^-^M group. Due to limited nutrition obtained from the mother in the egg, it subsequently leads to low hatching rate, pupation rate, emergence rate and longer developmental period. Our results provided evidence that endosymbionts impose a cost on their host under physiological stress.

## Figures and Tables

**Figure 1 insects-14-00558-f001:**
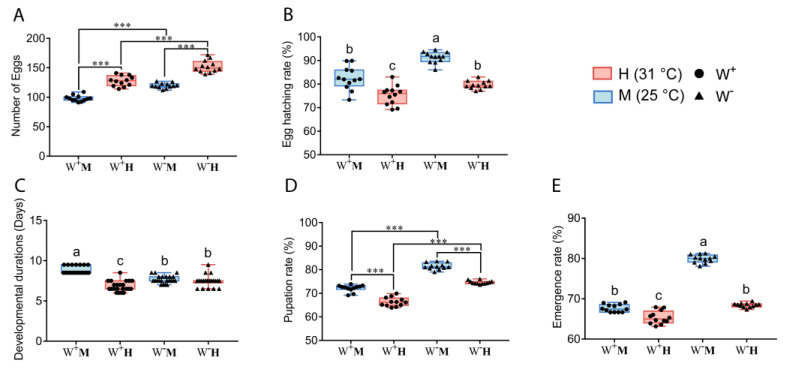
Effect of temperature and *Wolbachia* infection on the growth and development of F_1_ generation *D. melanogaster*. (**A**) Effect of temperature and *Wolbachia* on oviposition amount of *D. melanogaster*; (**B**) Effect of temperature and *Wolbachia* on hatching rate of *D. melanogaster*; (**C**) Effect of temperature and *Wolbachia* on developmental durations of *D. melanogaster*; (**D**) Effect of temperature and *Wolbachia* on pupation rate of *D. melanogaster*; (**E**) Effect of temperature and *Wolbachia* on emergence rate of *D. melanogaster*. Two-way ANOVA was used to analyze the interaction effect between temperature and *Wolbachia* infection on the F_1_ generation *D. melanogaster*. If there was an interaction between temperature and *Wolbachia* infection, Tukey HSD was used for multiple comparisons (*p* < 0.05), and different letters indicate significant differences between groups; if there was no interaction, Student’s *t*-test was used for pairwise comparisons (“***” indicates *p* < 0.001).

**Figure 2 insects-14-00558-f002:**
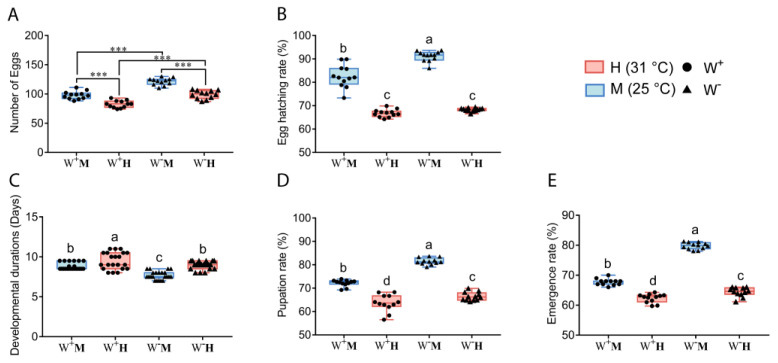
Effect of temperature and *Wolbachia* infection on the growth and development of F_2_ generation *D. melanogaster*. (**A**) Effect of temperature and *Wolbachia* on oviposition amount of *D. melanogaster*; (**B**) Effect of temperature and *Wolbachia* on hatching rate of *D. melanogaster*; (**C**) Effect of temperature and *Wolbachia* on developmental durations of *D. melanogaster*; (**D**) Effect of temperature and *Wolbachia* on pupation rate of *D. melanogaster*; (**E**) Effect of temperature and *Wolbachia* on emergence rate of *D. melanogaster*. Two-way ANOVA was used to analyze the interaction effect between temperature and *Wolbachia* infection on the F_2_ generation *D. melanogaster*. If there was an interaction between temperature and *Wolbachia* infection, Tukey HSD was used for multiple comparisons (*p* < 0.05), and different letters indicate significant differences between groups; if there was no interaction, Student’s *t*-test was used for pairwise comparisons (“***” indicates *p* < 0.001).

**Figure 3 insects-14-00558-f003:**
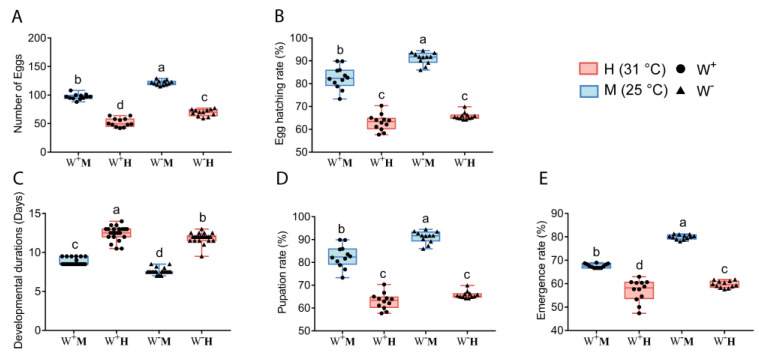
Effect of temperature and *Wolbachia* infection on the growth and development of F_3_ generation *D. melanogaster*. (**A**) Effect of temperature and *Wolbachia* on oviposition amount of *D. melanogaster*; (**B**) Effect of temperature and *Wolbachia* on hatching rate of *D. melanogaster*; (**C**) Effect of temperature and *Wolbachia* on developmental durations of *D. melanogaster*; (**D**) Effect of temperature and *Wolbachia* on pupation rate of *D. melanogaster*; (**E**) Effect of temperature and *Wolbachia* on emergence rate of *D. melanogaster*. Two-way ANOVA was used to analyze the interaction effect between temperature and *Wolbachia* infection on the F_3_ generation *D. melanogaster*. If there was an interaction between temperature and *Wolbachia* infection, Tukey HSD was used for multiple comparisons (*p* < 0.05), and different letters indicate significant differences between groups; if there was no interaction, Student’s *t*-test was used for pairwise comparisons.

**Figure 4 insects-14-00558-f004:**
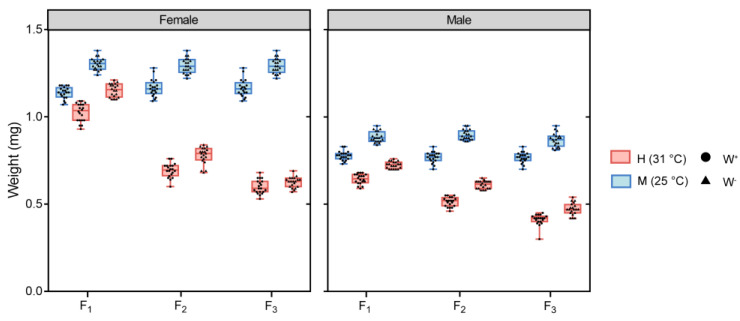
The comparisons of body weight of F_1,_ F_2_ and F_3_ generation *D. melanogaster* in W^+^M, W^-^M, W^+^H and W^-^H groups. Total n = 480.

**Figure 5 insects-14-00558-f005:**
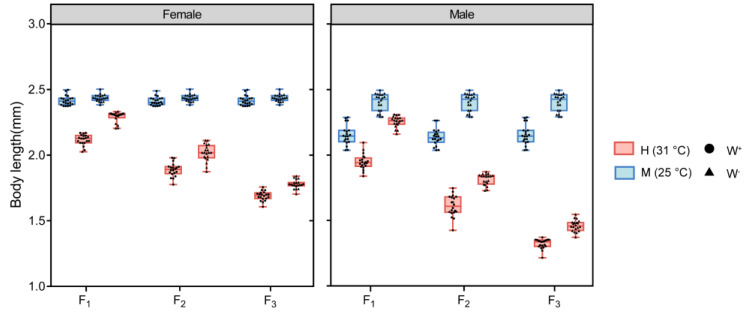
The comparisons of body length of F_1,_ F_2_ and F_3_ generation *D. melanogaster* in W^+^M, W^-^M, W^+^H and W^-^H groups. Total n = 480.

**Figure 6 insects-14-00558-f006:**
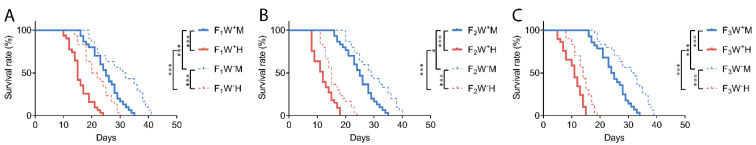
Effect of high temperature stress and *Wolbachia* infection on the survival rate of *D. melanogaster*. The survival curves were analyzed using the Kaplan–Meier method (log-rank test, * *p* < 0.05, *** *p* < 0.001). The solid line represents *Wolbachia* infection, the dashed line represents no *Wolbachia* infection, the red line represents fruit flies reared at 31 °C and the blue line represents fruit flies reared at 25 °C. (**A**) the survival curves of F_1_ generation *D. melanogaster*; (**B**) the survival curves of F_2_ generation *D. melanogaster*; (**C**) the survival curves of F_3_ generation *D. melanogaster*.

**Figure 7 insects-14-00558-f007:**
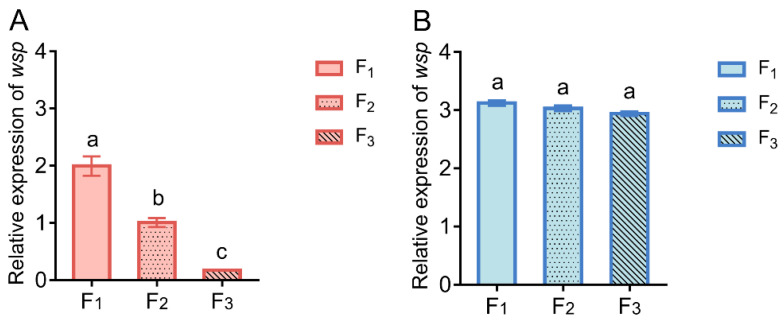
Effect of high temperature stress on *Wolbachia* infection. (**A**) Comparison of the relative expression of *wsp* among F_1_, F_2_ and F_3_ W^+^ flies reared at 31 °C. (**B**) Comparison of the relative expression of *wsp* among F_1_, F_2_ and F_3_ W^+^ flies reared at 31 °C (one-way ANOVA, *p* < 0.05, different lowercase letters indicate significant differences).

**Table 1 insects-14-00558-t001:** Effect of temperature and *Wolbachia* infection on the body weight of F_1,_ F_2_ and F_3_ generation *D. melanogaster*.

Effect	F_1_			F_2_			F_3_		
	df	F	*p*	df	F	*p*	df	F	*p*
**W infection**	**(1, 152)**	**501.639**	**<0.001**	**(1, 152)**	**347.281**	**<0.001**	**(1, 152)**	**176.253**	**<0.001**
**Temperature**	**(1, 152)**	**677.524**	**<0.001**	**(1, 152)**	**4182.671**	**<0.001**	**(1, 152)**	**6898.153**	**<0.001**
**Sex**	**(1, 152)**	**5474.552**	**<0.001**	**(1, 152)**	**2331.231**	**<0.001**	**(1, 152)**	**2369.755**	**<0.001**
**W infection × Temperature**	**(1, 152)**	**9.161**	**0.003**	**(1, 152)**	**9.559**	**0.002**	**(1, 152)**	**35.612**	**<0.001**
**W infection × Sex**	**(1, 152)**	**23.904**	**<0.001**	(1, 152)	0.352	0.554	(1, 152)	0.102	0.749
Temperature × Sex	(1, 152)	1.823	0.179	**(1, 152)**	**361.633**	**<0.001**	**(1, 152)**	**410.561**	**<0.001**
W infection × Temperature× Sex	(1, 152)	0.262	0.609	(1, 152)	0.303	0.583	**(1, 152)**	**6.652**	**0.011**

Note: Three-way ANOVA was used to analyze the interaction effect. Significant results are highlighted in bold.

**Table 2 insects-14-00558-t002:** Effect of temperature and *Wolbachia* infection on the body length of F_1,_ F_2_ and F_3_ generation *D. melanogaster*.

Effect	F_1_			F_2_			F_3_		
	df	F	*p*	df	F	*p*	df	F	*p*
**W infection**	**(1, 152)**	**501.639**	**<0.001**	**(1, 152)**	**347.281**	**<0.001**	**(1, 152)**	**176.253**	**<0.001**
**Temperature**	**(1, 152)**	**677.524**	**<0.001**	**(1, 152)**	**4182.671**	**<0.001**	**(1, 152)**	**6898.153**	**<0.001**
**Sex**	**(1, 152)**	**5474.552**	**<0.001**	**(1, 152)**	**2331.231**	**<0.001**	**(1, 152)**	**2369.755**	**<0.001**
**W infection × Temperature**	**(1, 152)**	**9.161**	**0.003**	**(1, 152)**	**9.559**	**0.002**	**(1, 152)**	**35.612**	**<0.001**
**W infection × Sex**	**(1, 152)**	**23.904**	**<0.001**	(1, 152)	0.352	0.554	(1, 152)	0.102	0.749
Temperature × Sex	(1, 152)	1.823	0.179	**(1, 152)**	**361.633**	**<0.001**	**(1, 152)**	**410.561**	**<0.001**
W infection × Temperature× Sex	(1, 152)	0.262	0.609	(1, 152)	0.303	0.583	**(1, 152)**	**6.652**	**0.011**

Note: Three-way ANOVA was used to analyze the interaction effect. Significant results are highlighted in bold.

## Data Availability

All data used in this paper are available within the text.

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
