# Peer review of "Interaction of High Temperature Stress and Wolbachia Infection on the Biological Characteristic of Drosophila melanogaster"

_insects, 2023, doi:10.3390/insects14060558_

Round 1

Reviewer 1 Report

Hu et al describe effects of temperature and Wolbachia infection on growth, development and survival rates of D. melanogaster.  I'm a little puzzled about the use of the word "interactive" in the title; perhaps the concept could be explained in the Introduction. Lines 277-278 in the Discussion likewise need to be explained more clearly. The simple summary is more reader friendly; too much detail is given in the abstract.

I have no real problems with the data as reported.  The main limitation is the small sample size, and whether the conclusions would hold with larger numbers of flies.  It would be interesting to synthesize an hypothesis from these data, and test it with a large sample.  Methods:  I would also like to see more detail on selection and monitoring of the Wolbachia-cured population, establishment of the isofemale line from wild flies, and selection of the 5-day old individuals (randomly, from how many; first emerged?)

These are a number of grammatical errors that could be corrected by a language editor. These include:

line 41, Gram

line 53, add common name for Hylyphantes

62: obligate;

needs careful proofreading throughout

Results: Replacing H and M with 31 and 25 would improve readability

Tables 1,2,3:  Use graphs instead of tables; are three significant figures justified by sample size?

Figure 3:  Specify the number of flies in these experiments

Line 309 to 310:  specify the fact that is supported by these results, so the reader doesn't have to look up the reference

In the context of evolution, it seems that there are two issues here:  how successfully do the flies transmit their own genes, and how successfully is the symbiont transmitted from one generation to the next, without being too deleterious to its host.  As Wolbachia is obviously abundant in insects, but in most cases not necessary, it would seem there has to be a fitness benefit to the infection, but these data suggest the opposite.  It would be interesting to see a more in-depth discussion of this issue.

These are a number of grammatical errors that could be corrected by a language editor. These include:

line 41, Gram

line 53, add common name for Hylyphantes

62: obligate;

needs careful proofreading throughout

Reviewer 2 Report

The paper by Hu et al. focuses on the effect of Wolbachia on Drosophila melanogaster under high temperature. In fact only one high temperature has been studied, 31°C (the authors do no justify this choice), and one Wolbachia strain (but the strain has not been indicated).

There are several points that need to be improved for this paper to be publishable.

Introduction :

The introduction is too short. There are three parts, the first one on Wolbachia, the second one focuses on temperature and the third one introduces the goals of the study. All the parts need to be more developed and there is no information on Drosophila melanogaster. For the Wolbachia part, there is a lack of information : the way Wolbachia is transmitted, the percentage of arthropods/nematods infected, what about Wolbachia in Drosophila melanogaster… Among Wolbachia effects, it is a shame that the nutritional role (in bed bugs for example) has not been indicated.

Moreover, there is a lack of bibliography. There are several papers that studied the interaction between Wolbachia and temperature on hosts (contrary to what is said l. 67/68), even on Drosophila melanogaster. The recent paper of Sturnov et al. (2022), that the authors quickly refered on in the discussion, must be cited in the introduction (an example among others… there is also the review paper of Corbin et al. in 2017…). And it should be interesting to have a more general view of the effect of symbiotic associations on hosts when they face to a thermal stress.

Material and methods :

There is a lack of information. For example : which Wolbachia strain is studied ? wMel or wMelCS ? what is the origin of the lines ? when and where the flies have been collected ? how and when the isofemale lines have been done ?

There are some other points that are questionable and/or must be clarified :

- l. 90. The number of eggs has been calculated on groups of 3 females. The problem with this protocol is when at least one female does not lay eggs. Moreover, are we sure that all the females are fertilized ?

- For the survival test, were the individuals fed ? Were they placed on the same medium as for the rearing ?

- l. 110-112. Please explain how F2 and F3 individuals have been obtained.

- For the Wolbachia density, it is weird that RNA and not DNA have been extracted. Usually this type of measurement is realised on DNA. Why do the authors chosed extracting RNA ? Have the measurements been done on males or females or both ? What was the age of individuals ? Sex and age have a great influence on Wolbachia density.

- l. 120. Please add the sequences of the actin primers.

- Please add details of the protocol used for determining Wolbachia density : quantity of DNA, concentration of primers, qPCR cyles…

- If I udestrand well, Wolbachia density have been measured on only 3 individuals/ modality/generation ? It is written that 5 individuals have been collected… or may be extractions have been made on poools of 5 individuals and 3 pools have been measured/modality/generation ? It is not clear at all, and, in my opinion, 3 measures are not enough.

- Part « data processing » : there is a big problem on statistical analyses. For example, I don’t understand how it is possible to perform student t tests on more than two modalities ! for comparing the 4 treatment combinations more complex statistics must been used, at least ANOVA2 if data are gaussian or general models. Moreover statistics used for testing differences on life history traits (for example survival) have not been indicated.

Results :

All this part must be rewritten after having done correct statistical analyses. Interactions between presence of Wolbachia and temperature must been tested first and if they are significant, specific effect of Wolbachia and temperature can’t be tested. ANOVA can not be used for analyses of survival data, other statistics such as Cox models are dedicated for this type of analysis.

Discussion :

- l. 251. Please precise. Data are avalaible.

- l. 255. Again, « is rarely studied » : it is not true. At least the authors must cite papers in which the same sort of study have been done (see review in Corbin et al. 2017 + more recent papers such as Sturnov et al. 2022).

- l. 270-276 : 6 lines without a dot ! Please rewrite.

- l. 299 to 310 : l. 301 : Does the term pathogen refered to Wolbachia ? because it is not the case, I dont understand the end of this discussion.

Please see my comments below:

- In the material and methods part, please

->change "cultured" by "reared"

-> change the term "selected". Individuals have not been selected but have been taken randomly, isn't it?

- l. 187 : table 1. it is not a "date the term more appropriate is "number" or "value"

-l. 188. it is written different temperature treatments, but in fact there is a control and one high temperature so only two temperatures...

Round 2

Reviewer 1 Report

These authors have responded adequately to reviewer concerns.  Proofreading for language is recommended.

These authors have responded adequately to reviewer concerns.  Proofreading for language is recommended.